# A Finite Element Model for the Vibration Analysis of Sandwich Beam with Frequency-Dependent Viscoelastic Material Core

**DOI:** 10.3390/ma12203390

**Published:** 2019-10-17

**Authors:** Zhicheng Huang, Xingguo Wang, Nanxing Wu, Fulei Chu, Jing Luo

**Affiliations:** 1College of Mechanical and Electrical Engineering, Jingdezhen Ceramic Institute, Jingdezhen 333001, China; huangzhicheng@jci.edu.cn (Z.H.); wunanxing@jci.edu.cn (N.W.); 2Department of Mechanical Engineering, Tsinghua University, Beijing 100084, China; chufl@mail.tsinghua.edu.cn; 3Beijing Research Institute of Automation for Machinery Industry Co., Ltd., Beijing 100120, China; sactc3@riamb.ac.cn

**Keywords:** sandwich beam, viscoelastic materials, frequency-dependent, finite elements, vibration analysis

## Abstract

In this work, a finite element model was developed for vibration analysis of sandwich beam with a viscoelastic material core sandwiched between two elastic layers. The frequency-dependent viscoelastic dynamics of the sandwich beam were investigated by using finite element analysis and experimental validation. The stiffness and damping of the viscoelastic material core is frequency-dependent, which results in complex vibration modes of the sandwich beam system. A third order seven parameter Biot model was used to describe the frequency-dependent viscoelastic behavior, which was then incorporated with the finite elements of the sandwich beam. Considering the parameters identification, a strategy to determine the parameters of the Biot model has been outlined, and the curve fitting results closely follow the experiment. With identified model parameters, numerical simulations were carried out to predict the vibration and damping behavior in the first three vibration modes, and the results showed that the finite model presented here had good accuracy and efficiency in the specific frequency range of interest. The experimental testing on the viscoelastic sandwich beam validated the numerical predication. The experimental results also showed that the finite element modeling method of sandwich beams that was proposed was correct, simple and effective.

## 1. Introduction

Viscoelastic materials have both elasticity and viscidity, and viscoelastic damping exists inside them, which can effectively suppress vibration and noise in engineering structures [1,2]. The elastic modulus of a viscoelastic material is too small to be used as a component alone. The widely used method in engineering is to bond it with an elastic material member to form viscoelastic sandwich composite structures. Figure 1 shows a typical viscoelastic sandwich composite beam structure. Due to the capacity to absorb shock, vibration suppression, and noise reduction, viscoelastic sandwich composite structures are widely used in the areas of aerospace, automobile, marine, many civil and mechanical applications. The viscoelastic core layer dissipates the energy through shear deformation due to its high damping capacity, and it is constrained by a stiffer elastic layer (called constraining layer) resulting in high transverse shear [3].

The dynamic properties of viscoelastic materials depend on many external factors, such as temperature, frequency, cyclic dynamic strain, static preloading, aging, radiation and oil pollution, etc. [4,5]. Among them, temperature and frequency have the greatest influence on the damping performance of viscoelastic materials. Many tests are required to determine the kinetic parameters. Usually these tests are conducted at isothermal conditions and only frequency dependence is taken into account. If the effects of temperature and frequency need to be considered simultaneously, the temperature–frequency superposition principle is applied to transform the influence of temperature into the influence of frequency [6]. Therefore, in the dynamic analysis of viscoelastic composite structures, the characteristics of viscoelastic materials varying with frequency needs to be mainly considered. 

The frequency-dependence of mechanical properties of viscoelastic materials leads to many difficulties in defining suitable mathematical models. Initially, the constitutive model of a viscoelastic material does not consider its frequency dependence. Early works on the damping characteristics of the viscoelastic composite structures were carried out by Kerwin Jr. [7], DiTaranto [8], Mead and Markus [9] and Yan and Dowell [10]. They developed analytical models to obtain approximate loss factors and natural frequencies of viscoelastic sandwich beams or plates. These studies used a complex-constant modulus model to characterize the mechanical properties of viscoelastic materials. However, they did not consider the frequency-dependent properties of viscoelastic materials. Later, some scholars proposed and used higher precision models. Park used the standard mechanical model to establish the analytical model of viscoelastic dampers [11]. Zhai [12] and Rezvani [13] derived an analytical model of the composite sandwich plates based on the first-order shear deformation theory. Manex presented an inverse method for the dynamic characterization of strong frequency-dependent viscoelastic materials from vibration test data [14]. Liu [15], Roman [16] and Lin [17] used a fractional order derivative model to describe the viscoelastic behavior of the viscoelastic materials. 

In engineering, the finite element method is commonly used to study the vibration of viscoelastic composite structures. Initially, people used solid elements in commercial finite element software to model viscoelastic composite structures [18,19,20]. These finite element methods used separate elements for each layer, resulting in a large amount of computation. To improve computational efficiency, a variety of composite finite elements have been developed. Chen, etc [21] established the integral finite elements for elastic-viscoelastic composite beams. Park, etc [22] compared two finite elements of viscoelastic composite plates. Daya and others [23] established a shell finite element for viscoelastically damped sandwich structures. However, they did not consider the frequency-dependent properties of viscoelastic materials. In order to consider the frequency-variation characteristics of viscoelastic material parameters in a finite element model, Adhikar and Manohar [24] used a stochastic finite-element formulation that employed frequency-dependent shape functions. Druesne et al. [25] used random fields to model the spatial variability of the mechanical properties of 3M ISD112 damping polymer. Following this, scholars studied the combination of these finite element methods and viscoelastic constitutive models. One of the more effective viscoelastic damping models in engineering applications is the Golla–Hughes–McTavish (GHM) model. It was developed by Golla and Hughes [26] and McTavish and Hughes [27,28]. By introducing auxiliary coordinates, the GHM model can be incorporated with the finite element dynamic equation of viscoelastic composite structures. Many scholars have used this method to study the vibration of viscoelastic composite structures [29,30,31]. In addition, Lesieutre and Lee introduced the Anelastic Displacement Fields (ADF) model [32,33]. Both GHM and ADF methods were studied and compared by Wang [34]. The Biot [35] model was first proposed in 1955. It was originally used to study the irreversible thermodynamic behavior of viscoelastic materials. However, the model has not received sufficient attention in the modeling of the viscoelastic composite structures for a long time. The introduction and research on parameter determination and the engineering application of the Biot model is rare. From these references it can be observed that the fractional derivative model has a large amount of calculations when performing vibration analysis of viscoelastic composite structures, and the GHM model can lead to a high system matrix order. The lack of a Biot model to analyse the viscoelastic composite structures, and those specifically in the finite element method, are both the principal motivation for the present study.

In this work, we have developed and proposed a simple and efficient finite element method incorporated with the Biot model for the vibration analysis of viscoelastic composite structures. First, according to the constitutive equation of the viscoelastic material Biot model, an effective method for determining the parameters of the Biot model is described in detail. Then, a method combining the Biot model with the general finite element dynamic equation of the viscoelastic sandwich beam is given. The derivation of the finite element equation follows the Euler–Bernoulli beam theory. It was assumed that there was no relative sliding between the layers, and the influence of the moment of inertia was not counted. Finally, the method was verified by experiments.

## 2. Finite Element Methods for a Sandwich Beam with a Frequency-Dependent Viscoelastic Material

### 2.1. Viscoelastic Models Using Biot Methods

The mechanical simulation of the Biot model is shown in Figure 2 [35]. As can be seen from Figure 2, the Biot model parallels a series of Maxwell model elements (also known as mini-oscillator), and then parallels a spring. The mini-oscillators were coupled with the spatial coordinates of the system by the auxiliary dissipative coordinate “Z” to simulate the stress–strain behavior of the viscoelastic material corresponding to the displacement.

Then, the relaxation function of the viscoelastic material can be expressed as:(1)G(t)=G∞+∑i=1NG∞aietbi
when performing the Laplace transform, the expression of the Biot model can be obtained as:(2)$$s\tilde{G}(s)=G^{\infty }\left[1+\sum_{i=1}^{N}\frac{a_{i}s}{s+b_{i}}\right]$$
when s=jω, one can obtain the curve fitting expression in the frequency domain:(3)$$G^{*}(j\omega )=G^{\infty }\left[1+\sum_{i=1}^{N}\frac{a_{i}\left(jw\right)}{\left(jw\right)+b_{i}}\right]$$
where $G^{\infty }$ is the equilibrium (steady state) value of the shear modulus of the viscoelastic material,$\left\{a_{k},b_{k}\right\}$ are positive constants, *i* = 1,2,3,…,*N,* and *N* is the number of micro-vibrator series. If N-order micro-vibrator is taken, the model has 2N+1 parameters to be determined. These parameters can be obtained by nonlinear curve fitting of experimental data.

### 2.2. Assumptions of a Sandwich Beam

In order to facilitate sandwich beam modeling with viscoelastic damping treatments, the following assumptions were made:(1)The constraining layer and the base beam satisfy two hypotheses of the Euler–Bernoulli beam theory, that is, the section perpendicular to the centerline of the beam is still planar after deformation (assumption of rigid cross section) and after deformation, the plane of the cross section is still perpendicular to the deformed axis.(2)Regardless of the vertical transverse compression deformation, the base beam layer, the damping layer, and the constraining layer is considered to have the same deflection.(3)The influence of moment of inertia is negligible relative to bending and tensile deformation of the elastic layers and the shear deformation of the viscoelastic layer. Therefore, in order to simplify the modeling process, the moment of inertia is ignored.(4)The layers of the materials are firmly bonded and there is no relative sliding between the layers.

### 2.3. Kinematics

The geometry and deformation relationship of each layer of the sandwich beam is shown in Figure 3. The dotted line in the figure is the middle face of each layer, and the meaning of each geometric quantity is as follows: $h_{1}$,$h_{2}$ and $h_{3}$ is the thickness of the base beam, the viscoelastic layer, and the constraining layer, respectively; w and $\frac{\partial w}{\partial x}$ is the transverse deflection and rotation angle of the sandwich beam, respectively; $u_{1}$,$u_{2}$ and $u_{3}$ is the longitudinal (x-direction) displacement of the base beam, the viscoelastic layer, and the constraining layer, respectively; φ and β is the rotation angle (shear angle) and the shear strain of the viscoelastic layer, respectively; d is the distance of the centerlines between the constraining layer and the base beam.

According to the assumptions and the First Order Shear Deformation Theory (FSDT), the displacements of any point in the two elastic skin layers of the sandwich beam in the right-handed coordinate system can be written as [20]:(4)$$\begin{array}{l} u^{\left(i\right)}(x,z,t)=u_{i}(x,t)-z_{i}\frac{\partial w_{i}(x,t)}{\partial x}\\ w^{\left(i\right)}(x,z,t)=w_{i}(x,t) \end{array}\}\;i=1,3$$
where $u^{\left(i\right)}$ and $w^{\left(i\right)}$ are the longitudinal and transverse displacements of the neutral surface of the *i*th face layer, respectively; $z_{i}$ is the distance from the neutral surface of the *i*th face layer.

It can be seen from Figure 3 that at the top surface of the viscoelastic layer (the bottom surface of the constraining layer), the z-coordinate is $z_{3}=-h_{3}/2$, and at the bottom surface of the viscoelastic layer (the top surface of the base beam), the z-coordinate is $z_{1}=h_{1}/2$. Substituting them into the first formula of Equation (4), respectively, the x-direction displacement of the top and bottom surface of the viscoelastic layer can be expressed as
(5)$$u_{top}=u_{3}+\frac{h_{3}}{2}\frac{\partial w}{\partial x},\;u_{bot}=u_{1}-\frac{h_{1}}{2}\frac{\partial w}{\partial x}$$

Then, the x-direction displacement of the viscoelastic layer can be expressed as:(6)$$u_{2}=\frac{u_{top}+u_{bot}}{2}$$

Substituting Equation (5) into Equation(6), the mid-plane displacement of the viscoelastic layer along the x direction takes the form
(7)$$u_{2}=\frac{1}{2}\left[\left(u_{3}+u_{1}\right)+\left(\frac{h_{3}-h_{1}}{2}\right)\frac{\partial w}{\partial x}\right]$$

According to the geometric relationship, the shear angle of the viscoelastic layer around the *y*-axis is given by
(8)$$\varphi =\frac{u_{top}-u_{bot}}{h_{2}}.$$

Substituting Equation (5) into Equation (8) gives
(9)$$\varphi =\frac{1}{h_{2}}\left[\left(u_{3}-u_{1}\right)+\frac{h_{3}+h_{1}}{2}\frac{\partial w}{\partial x}\right]$$

The shear strain of the viscoelastic layer can be expressed as
(10)$$\beta =\frac{\partial w}{\partial x}+\varphi$$

Substituting Equation (9) into Equation (10), one has
(11)$$\beta =\frac{1}{h_{2}}\left[\left(u_{3}-u_{1}\right)+d\frac{\partial w}{\partial x}\right]$$
where $d=\frac{h_{3}+h_{1}}{2}+h_{2}$ is the distance of the centerlines between the constraining layer and the base beam.

### 2.4. Degrees of Freedom and Shape Functions

The nodal degrees of freedom (DOF) of the finite element that was to be developed is shown in Figure 4. It was a 2-node 8-degree-of-freedom composite beam element with a length of $l_{e}$ and a width of b. The nodal degrees (four DOF of each node) are represented by the longitudinal displacement $u_{1}$ and $u_{3}$, the transverse displacement w, and the rotation angle θ=∂w/∂x.

The nodal displacement vector is given by
(12)$$\Delta^{e}={\left\{\begin{array}{cccccccc} u_{1i} & u_{3i} & w_{i} & \theta_{i} & u_{1j} & u_{3j} & w_{j} & \theta_{j} \end{array}\right\}}^{T}$$


The displacement of any point in the element can be determined by the displacement of the 8 nodes of the element by the shape function interpolation, as follows
(13)$$\Delta ={\left[\begin{array}{cccc} u_{1} & u_{3} & w & \theta \end{array}\right]}^{T}=N\Delta^{e}$$
where $N=\left[\begin{array}{cccc} N_{1} & N_{2} & N_{3} & N_{4} \end{array}\right]$ is the shape function matrix corresponding to the four displacement components of the element, which is a 4×8 form, and its component expressions are:(14)$$N_{1}=\left[\begin{array}{cccccccc} \left(1-\xi \right) & 0 & 0 & 0 & \xi & 0 & 0 & 0 \end{array}\right]$$
(15)$$N_{2}=\left[\begin{array}{cccccccc} 0 & \left(1-\xi \right) & 0 & 0 & 0 & \xi & 0 & 0 \end{array}\right]$$
(16)$$N_{3}=\left[\begin{array}{cccccccc} 0 & 0 & \left(1-3\xi^{2}+2\xi^{3}\right) & \left(\xi -2\xi^{2}+\xi^{3}\right)l_{e} & 0 & 0 & \left(3\xi^{2}-2\xi^{3}\right) & (-\xi^{2}+\xi^{3})l_{e} \end{array}\right]$$
(17)$$N_{4}=\left[\begin{array}{cccccccc} 0 & 0 & \frac{6\xi \left(\xi -1\right)}{l_{e}} & \left(1-4\xi +3\xi^{2}\right) & 0 & 0 & \frac{6\xi (1-\xi )}{l_{e}} & \left(-2\xi +3\xi^{2}\right) \end{array}\right]$$
where $\xi =x/l_{e}$ is a dimensionless coordinate.

Then, the four displacement components of the finite element represented by the shape function can be obtained as
(18)$$u_{1}=N_{1}\Delta^{e},u_{3}=N_{2}\Delta^{e},w=N_{3}\Delta^{e},\theta =N_{4}\Delta^{e}$$

Substituting Equation (18) into Equation (7) gives
(19)$$u_{2}=\frac{1}{2}\left[\left(N_{1}+N_{2}\right)+\left(\frac{h_{3}-h_{1}}{2}\right)N_{4}\right]\Delta^{e}=N_{5}\Delta^{e}$$
where
(20)$$N_{5}=\frac{1}{2}\left[\left(N_{1}+N_{2}\right)+\left(\frac{h_{3}-h_{1}}{2}\right)N_{4}\right]$$


Substituting Equation (18) into Equation (11) gives
(21)$$\beta =\frac{1}{h_{2}}\left[\left(N_{1}-N_{2}\right)+\left(\frac{h_{1}+h_{3}}{2}+h_{2}\right)N_{4}\right]\Delta^{e}=N_{6}\Delta^{e}$$
where
(22)$$N_{6}=\frac{1}{h_{2}}\left[\left(N_{1}-N_{2}\right)+\left(\frac{h_{1}+h_{3}}{2}+h_{2}\right)N_{4}\right]$$


### 2.5. Energy Terms

To establish the dynamic equation of the sandwich beam element, it is necessary to use the energy relationship of the structure to derive its mass and stiffness matrixes. System energy includes strain energy and kinetic energy. The nomenclature used in this section is presented as follows: $E_{i},A_{i},I_{i}$ and $\rho_{i}\;$
(i=1,2,3) are Yong’s modulus, the cross-sectional area, the moment of inertia and the density of the base beam, the viscoelastic layer and the constraining layer, respectively. $G_{2}$ is the shear modulus of the viscoelastic layer.

#### 2.5.1. The Strain Energy

The strain energy accounting for extensional, bending and shear effects can be given by
(23)$$U=U_{e1}+U_{b1}+U_{e3}+U_{b3}+U_{s2}$$
where U is the overall strain energy, $U_{e1}$ and $U_{e3}$ are the extensional strain energy of the base beam and the constraining layer, respectively; $U_{b1}$ and $U_{b3}$ are the bending strain energy of the base beam and the constraining layer, respectively; $U_{s2}$ is the shear strain energy of the viscoelastic layer. The expressions of these strain energies are as follows
(24)$$U_{e1}=\frac{1}{2}E_{1}A_{1}{\int_{0}^{l_{e}}\left(\frac{\partial u_{1}}{\partial x}\right)}^{2}dx=\frac{1}{2}\Delta^{e}{}^{{}^{T}}K_{e1}^{e}\Delta^{e}\;U_{b1}=\frac{1}{2}E_{1}I_{1}{\int_{0}^{l_{e}}\left(\frac{\partial^{2}w}{\partial x^{2}}\right)}^{2}dx=\frac{1}{2}\Delta^{e}{}^{{}^{T}}K_{b1}^{e}\Delta^{e}\;U_{e3}=\frac{1}{2}E_{3}A_{3}{\int_{0}^{l_{e}}\left(\frac{\partial u_{3}}{\partial x}\right)}^{2}dx=\frac{1}{2}\Delta^{e}{}^{{}^{T}}K_{e3}^{e}\Delta^{e}\;U_{b3}=\frac{1}{2}E_{3}I_{3}{\int_{0}^{l_{e}}\left(\frac{\partial^{2}w}{\partial x^{2}}\right)}^{2}dx=\frac{1}{2}\Delta^{e}{}^{{}^{T}}K_{b3}^{e}\Delta^{e}\;U_{s2}=\frac{1}{2}G_{2}A_{2}\int_{0}^{l_{e}}\beta^{2}dx=\frac{1}{2}\Delta^{e}{}^{{}^{T}}K_{v}^{e}\Delta^{e}$$
where $K_{e1}^{e}$ and $K_{b1}^{e}$ are the stiffness matrices of the base beam corresponding to extension and bending, respectively; $K_{e3}^{e}$ and $K_{b3}^{e}$ are the stiffness matrices of the constraining layer corresponding to extension and bending, respectively; $K_{v}^{e}$ is the stiffness matrix of the viscoelastic layer corresponding to shear. Applying the shape function, and the expressions of these stiffness matrices were as follows
(25)$$K_{e1}^{e}=E_{1}A_{1}l_{e}{\int_{0}^{1}\left[\frac{\partial N_{1}}{\partial x}\right]}^{T}\left[\frac{\partial N_{1}}{\partial x}\right]d\zeta \;K_{b1}^{e}=E_{1}I_{1}l_{e}{\int_{0}^{1}\left[\frac{\partial^{2}N_{3}}{\partial x^{2}}\right]}^{T}\left[\frac{\partial^{2}N_{3}}{\partial x^{2}}\right]d\zeta \;K_{e3}^{e}=E_{3}A_{3}l_{e}{\int_{0}^{1}\left[\frac{\partial N_{2}}{\partial x}\right]}^{T}\left[\frac{\partial N_{2}}{\partial x}\right]d\zeta \;K_{b3}^{e}=E_{3}I_{3}l_{e}{\int_{0}^{1}\left[\frac{\partial^{2}N_{3}}{\partial x^{2}}\right]}^{T}\left[\frac{\partial^{2}N_{3}}{\partial x^{2}}\right]d\zeta \;K_{v}^{e}=G_{2}A_{2}l_{e}\int_{0}^{1}N_{6}{}^{T}N_{6}d\zeta$$

The total stiffness matrix is the sum of the three layers of stiffness and it is written as
(26)$$K^{e}=\underset{K_{e}^{e}}{\underset{︸}{K_{ec}^{e}+K_{bc}^{e}+K_{eb}^{e}+K_{bb}^{e}}}+K_{v}^{e}$$
where $K_{e}^{e}$ is the elastic stiffness matrix, which is the sum of the first four terms on the right side of the equation, and $K_{v}^{e}$ is the viscous stiffness matrix.

#### 2.5.2. The Kinetic Energy

The kinetic energy including longitudinal and transverse motion yields
(27)$$T=T_{e1}+T_{b1}+T_{e2}+T_{b2}+T_{e3}+T_{b3}$$
where T is the overall kinetic energy; $T_{e1}$, $T_{e2}$ and $T_{e3}$ are the kinetic energies of the base beam, the constraining layer and the constraining layer associated with longitudinal motion, respectively; $T_{b1}$, $T_{b2}$ and $T_{b3}$ are the kinetic energies of the base beam, the constraining layer and the constraining layer associated with transverse motion, respectively. The expressions of these kinetic energies were as follows
(28)$$T_{e1}=\frac{1}{2}\rho_{1}A_{1}{\int_{0}^{l_{e}}\left(\frac{\partial u_{1}}{\partial t}\right)}^{2}dx=\frac{1}{2}{\dot{\Delta }}^{e}{}^{{}^{T}}M_{e1}^{e}{\dot{\Delta }}^{e}\;T_{b1}=\frac{1}{2}\rho_{1}A_{1}{\int_{0}^{l_{e}}\left(\frac{\partial w}{\partial t}\right)}^{2}dx=\frac{1}{2}{\dot{\Delta }}^{e}{}^{{}^{T}}M_{b1}^{e}{\dot{\Delta }}^{e}\;T_{e2}=\frac{1}{2}\rho_{2}A_{2}{\int_{0}^{l_{e}}\left(\frac{\partial u_{2}}{\partial t}\right)}^{2}dx=\frac{1}{2}{\dot{\Delta }}^{e}{}^{{}^{T}}M_{e2}^{e}{\dot{\Delta }}^{e}\;T_{bb}=\frac{1}{2}\rho_{2}A_{2}{\int_{0}^{l_{e}}\left(\frac{\partial w}{\partial t}\right)}^{2}dx=\frac{1}{2}{\dot{\Delta }}^{e}{}^{{}^{T}}M_{b2}^{e}{\dot{\Delta }}^{e}\;T_{e3}=\frac{1}{2}\rho_{3}A_{3}{\int_{0}^{l_{e}}\left(\frac{\partial u_{3}}{\partial t}\right)}^{2}dx=\frac{1}{2}{\dot{\Delta }}^{e}{}^{{}^{T}}M_{e3}^{e}{\dot{\Delta }}^{e}\;T_{b3}=\frac{1}{2}\rho_{3}A_{3}{\int_{0}^{l_{e}}\left(\frac{\partial w}{\partial t}\right)}^{2}dx=\frac{1}{2}{\dot{\Delta }}^{e}{}^{{}^{T}}M_{b3}^{e}{\dot{\Delta }}^{e}$$
where $M_{e1}^{e}$, $M_{b1}^{e}$, $M_{e2}^{e}$, $M_{b2}^{e}$, $M_{e3}^{e}$ and $M_{b3}^{e}$ are the mass matrices of the base beam, the constraining layer and the constraining layer associated with longitudinal motion and transverse motion, respectively. Applying the shape function and the expressions of these mass matrixes were as follows
(29)$$M_{e1}^{e}=\rho_{1}A_{1}l_{e}\int_{0}^{1}N_{1}^{T}N_{1}d\zeta \;M_{b1}^{e}=\rho_{1}A_{1}l_{e}\int_{0}^{1}N_{3}^{T}N_{3}d\zeta \;M_{e2}^{e}=\rho_{2}A_{2}l_{e}\int_{0}^{1}N_{2}^{T}N_{2}d\zeta \;M_{b2}^{e}=\rho_{2}A_{2}l_{e}\int_{0}^{1}N_{3}^{T}N_{3}d\zeta \;M_{e3}^{e}=\rho_{3}A_{3}l_{e}\int_{0}^{1}N_{5}^{T}N_{5}d\zeta \;M_{b3}^{e}=\rho_{v}A_{v}l_{e}\int_{0}^{1}N_{3}^{T}N_{3}d\zeta$$

The total mass matrix is the sum of the three layers of mass matrices and it was written as
(30)$$M^{e}=M_{e1}^{e}+M_{b1}^{e}+M_{e2}^{e}+M_{b2}^{e}+M_{e3}^{e}+M_{b3}^{e}$$

### 2.6. Equation of Motion with Viscoelastic Damping

The viscoelastic behavior of the sandwich structure was decomposed by an elastic part and an anelastic part. Thus, the finite element dynamic equation of the viscoelastic composite structure can be expressed in the Laplace domain as [10,11]:(31)$$\left(s^{2}M^{e}+K_{e}{}^{e}+G^{*}\left(s\right)K_{{}_{v}}^{e}\right)x\left(s\right)=f^{e}\left(s\right)$$
where $M^{e}$ is the total mass matrix of the element including elastic and viscoelastic materials, $K_{e}{}^{e}$ is the stiffness matrix of elastic materials, $K_{v}^{e}$ is the complex stiffness matrix of viscoelastic materials, x(s) is the displacement vector, $f^{e}$ is the excitation vector. $G^{*}(s)=s\tilde{G}(s)$ is the complex modulus of viscoelastic materials.

Substituting the Biot model Equation (2) into Equation (31), and then introducing an auxiliary coordinate ${\hat{Z}}_{k}\left(s\right)$, which was defined as
(32)$$\left\{{\hat{Z}}_{k}\left(s\right)\right\}=\frac{b_{k}}{s+b_{k}}\left\{x\left(s\right)\right\}$$
where k=1,2,3⋅⋅⋅N, one can obtain the equation of motion: (33)$$\bar{M}\ddot{q}+\bar{D}\dot{q}+\bar{K}q=\bar{f}$$

Here each matrix and vector was defined as follows
(34)$$\bar{M}=\left[\begin{array}{cccc} M^{e} & 0 & \cdots & 0\\ 0 & 0 & \cdots & 0\\ \vdots & \vdots & \ddots & \vdots \\ 0 & 0 & \cdots & 0 \end{array}\right]\;\bar{D}=\left[\begin{array}{cccc} 0 & 0 & \cdots & 0\\ 0 & \frac{a_{1}}{b_{1}}\Lambda & \cdots & 0\\ \vdots & \vdots & \ddots & \vdots \\ 0 & 0 & \cdots & \frac{a_{N}}{b_{N}}\Lambda \end{array}\right]\;\bar{K}=\left[\begin{array}{cccc} K_{e}{}^{e}+\tilde{k}\left(1+\sum_{K=1}^{N}a_{k}\right) & -a_{1}R & \cdots & -a_{N}R\\ -a_{1}R^{T} & a_{1}\Lambda & \cdots & 0\\ \vdots & \vdots & \ddots & \vdots \\ -a_{N}R^{T} & 0 & \cdots & a_{N}\Lambda \end{array}\right]\;q=\left\{\begin{array}{c} x\\ Z_{1}\\ \vdots \\ Z_{N} \end{array}\right\}\;\bar{f}=\left\{\begin{array}{c} f\\ 0\\ \vdots \\ 0 \end{array}\right\}$$
where $\tilde{k}=G^{\infty }K_{v}^{e}$, $K_{v}^{e}=R_{v}\Lambda_{v}R_{v}^{T}$, $\Lambda_{v}$ is a diagonal matrix composed of the positive eigenvalues of the viscoelastic stiffness matrix $K_{v}^{e}$,$R_{v}$ is a matrix with corresponding orthogonal eigenvectors as columns, $\Lambda =G^{\infty }\Lambda_{v}$, $R=R_{v}\Lambda$, $Z_{j}=R_{{}_{v}}^{T}{\hat{Z}}_{j}$, (j=1,2,⋯,N).

According to the general element integration method in the finite element method, the physical coordinate X of the viscoelastic sandwich beam structure is integrated, and the corresponding boundary conditions are introduced to obtain the overall dynamic equation as follows:(35)$$M\ddot{x}+D\dot{x}+Kx=F$$
where M,D,K are the total mass matrix, damping matrix and stiffness matrix of the sandwich beam structure, respectively. F is the motivating force for the system.

Obviously, Equation (35) is a common second-order constant-linear system dynamic equation. It is very convenient to solve the modal parameters such as natural frequency and damping. It can also directly use the linear system control theory to actively control the vibration of viscoelastic composite structures. The advantages make the Biot model valuable as a good engineering application.

## 3. Numerical Simulation and Experimental Validation

### 3.1. Curve Fits for Biot Model Parameter

In order to accurately determine the frequency dependent viscoelastic behavior of the viscoelastic material, the Biot model parameters were evaluated through curve fits of viscoelastic modulus models with experimental measurement at 30 °C.

The viscoelastic material used in the experiment was ZN-1 viscoelastic material developed by the China Institute of Aerospace Materials and Technology. This material is especially suitable for constructing sandwich structure and is widely used. The ZN-1 viscoelastic material is made into standard test pieces as shown in Figure 5, which are cylinders having a diameter and a height of 20 mm. A dynamic viscoelastic spectrometer (as shown in Figure 6, VISCOANALYSEUR VA4000 manufactured by France METRAVIB) was used to measure the storage modulus and loss factor of the viscoelastic material at different excitation frequencies. The temperature at the time of measurement was set to 30°C. This data has been listed in Table 1.

The imaginary part and the real part of the Equation (3) are called the loss modulus and the storage modulus of the viscoelastic material, respectively, and the ratio of them is the loss factor. In order to determine the Biot model parameters in Equation (3), the following optimization objective function was established:(36)$$F(x)=\sum_{i=1}^{M}{\left|G^{*}\left(x,\omega_{i}\right)-G_{0}\left(\omega_{i}\right)\right|}^{2}=\min$$
where, $G^{*}\left(x,\omega_{i}\right)$ is the Biot model expression with parameters to be determined, $G_{0}\left(\omega_{i}\right)$ is the measured complex modulus value in the complex frequency domain. The real part of $G_{0}\left(\omega_{i}\right)$ is the storage modulus in Table 1, and the imaginary part is the product of the storage modulus and the loss factor in Table 1. M is the number of measured complex modulus, x is the parameter of the Biot model to be determined, and its expression was:(37)$$x_{1}=G^{\infty };x_{2}=a_{1},x_{3}=a_{2},\cdot \cdot \cdot x_{N+1}=a_{N};x_{N+2}=b_{1},x_{N+3}=b_{2},\cdot \cdot \cdot x_{2N+1}=b_{N}$$

This is a nonlinear optimization problem with constraints in the complex plane. The optimal constraint was:(38)$$x_{i}>0,i=1,2,3\cdot \cdot \cdot 2N+1$$

Based on the experimental data of Table 1, the parameters of the Biot model could be obtained by solving the aforementioned optimization problem. Figure 7 and Figure 8 illustrate the fitted storage modulus (the real part) and the loss modulus (imaginary part) using the Biot model compared with experimental data in Table 1, respectively. Figure 9 shows the error.

The Biot model parameters identified by curve fitting are presented in Table 2. 

It can be seen from Figure 7 and Figure 8 that when the three micro vibrators were used, the fitted Biot model could approximate the experimental values of the viscoelastic material well. It truly reflected the variation of the viscoelastic material parameters with frequency. It can be seen from Figure 9 that the error near 5 Hz was large at the beginning of the experiment. This was because when using a dynamic viscoelastic spectrometer, a vibration must be applied to the test piece, and low vibration frequency would reduce the accuracy of the test. When the frequency was increased, the fitting error significantly reduced. When the frequency was in the range of 10 Hz∼500 Hz, the error of the real and imaginary part were less than 3%. Moreover, the high frequency range was better than the low frequency, the imaginary part was better than the real part. Therefore, the Biot model could near perfectly fit the experimental data. That is to say, the parameter identification method of the Biot model presented here was effective.

### 3.2. Experimental Validation of the Sandwich Composite Cantilever Beam

The aforementioned viscoelastic Biot model was incorporated into the finite element model of the sandwich composite cantilever beam presented in Figure 1. The material and mechanical properties are given in Table 3. 30 finite elements were used to discrete the sandwich beam. In order to verify the numerical simulation results of the presented finite element method, a free vibration response was experimentally tested to obtain the first three natural frequencies and loss factors of the beam. The experimental device is shown in Figure 10. The experimental and the finite element calculation results are listed in Table 4.

It can be seen from Table 4 that the calculation results of the finite element model in this paper were in good agreement with the experimental results. The prediction error of the natural frequency corresponding to the first three modes of the viscoelastic sandwich beam was less than 4%, and the prediction error corresponding to the first three-order loss factor was below 5%. This showed that the method in this paper was correct and effective.

The uncertainties, the repeativity and reproducibility of measurements should be considered in the experimental process. The uncertainties of the measurement were “the parameters that characterize the dispersion of the values reasonably assigned to the measurements and is related to the results of the measurements”. The repeativity of measurements was the degree of consistency between a series of results obtained under the same conditions using the same method and the same test material. The reproducibility of measurements was the degree of consistency between individual results obtained under different conditions, using the same method and the same test material. During the experiment, the same test piece was measured five times by two operators, respectively and then a total of 10 results were averaged as the measurement results. The results showed that the uncertainties, the repeativity and reproducibility of measurements were all within an acceptable range.

## 4. Conclusions

Based on the first-order shear deformation theory and Hamilton principle, a general finite element model of viscoelastic sandwich beams was established. A 2-node 8-DOF composite beam element was established for structural discretization of viscoelastic sandwich beams. The Biot constitutive model was used to consider the frequency-dependent properties of a viscoelastic material. A method for determining its parameters and incorporating it with the finite element equation of viscoelastic composite beam structure was proposed. The finite element equation of the viscoelastic sandwich beam structure was transformed into a common second-order constant linear system dynamic equation to improve the efficiency of the solution. Considering the parameter identification, curve fits of the Biot model compared with experimental data were presented, and a good agreement (less than 5 percent error) was reached. A numerical simulation was carried out to predict the damping behavior of a viscoelastic sandwich cantilever beam in its first three vibration modes by using the finite element model presented in this paper. The experimental testing on the viscoelastic sandwich cantilever beam validated the numerical predication pretty well. The results showed that the finite element model in this paper had good calculation accuracy and efficiency, and had a good engineering popularization value.

## Figures and Tables

**Figure 1 materials-12-03390-f001:**
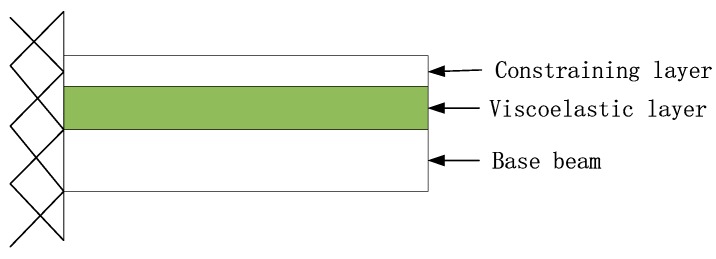
Viscoelastic composite beam structure.

**Figure 2 materials-12-03390-f002:**
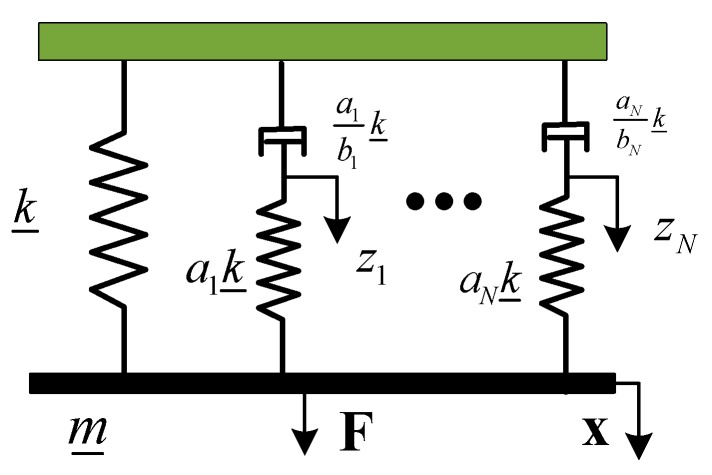
Mechanical analogy of the Biot model.

**Figure 3 materials-12-03390-f003:**
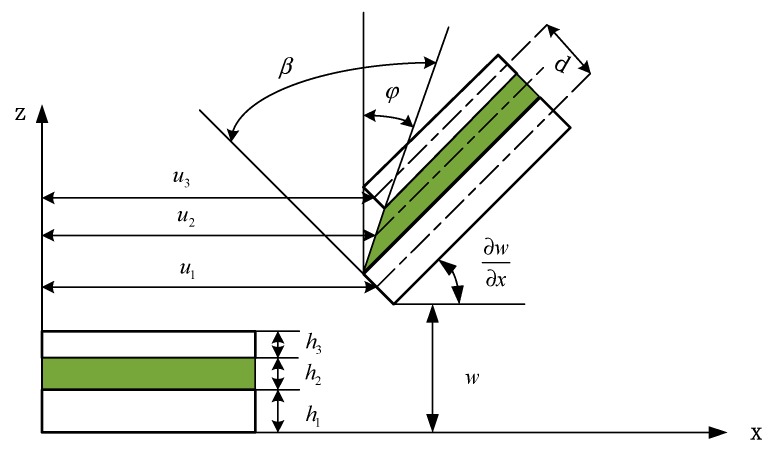
Geometry and deformation of sandwich beam.

**Figure 4 materials-12-03390-f004:**
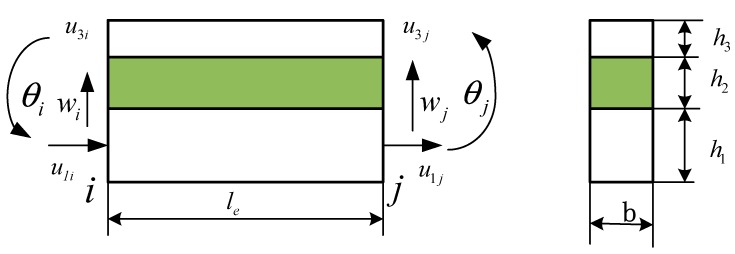
The element of sandwich beam.

**Figure 5 materials-12-03390-f005:**
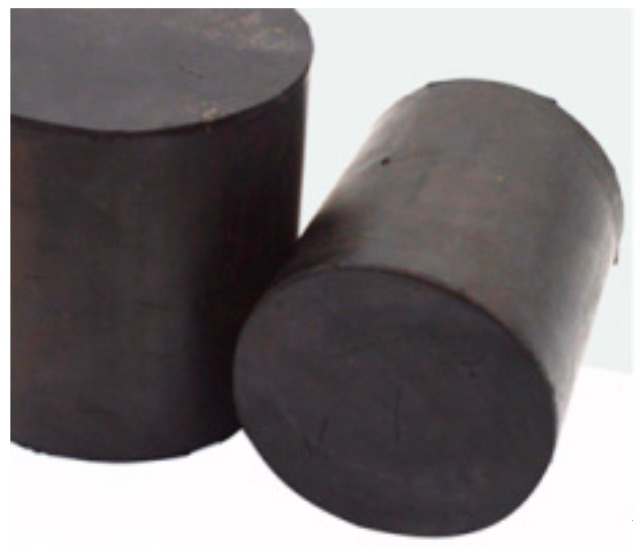
The ZN-1 viscoelastic material standard test piece.

**Figure 6 materials-12-03390-f006:**
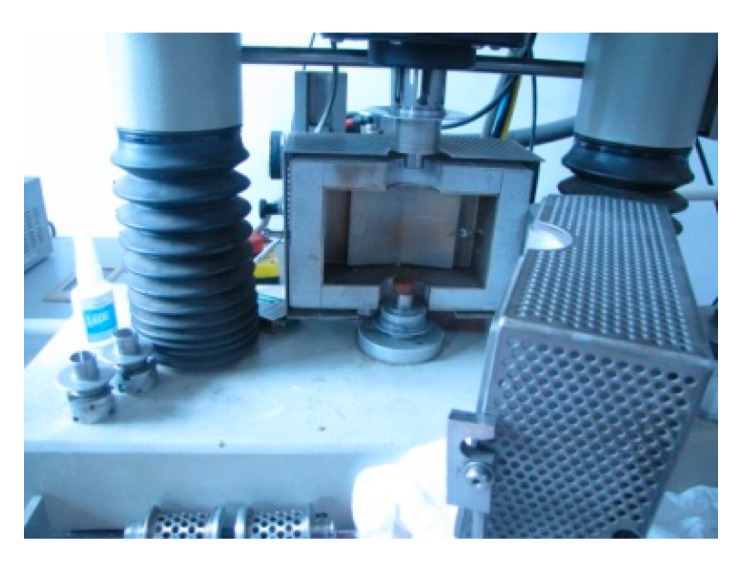
VISCOANALYSEUR VA4000 dynamic viscoelastic spectrometer.

**Figure 7 materials-12-03390-f007:**
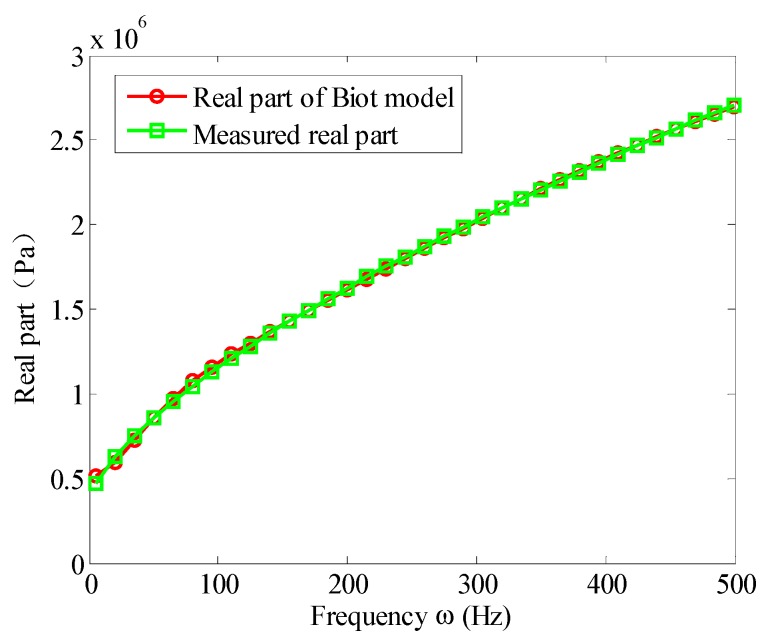
Real part fitting of the Biot model.

**Figure 8 materials-12-03390-f008:**
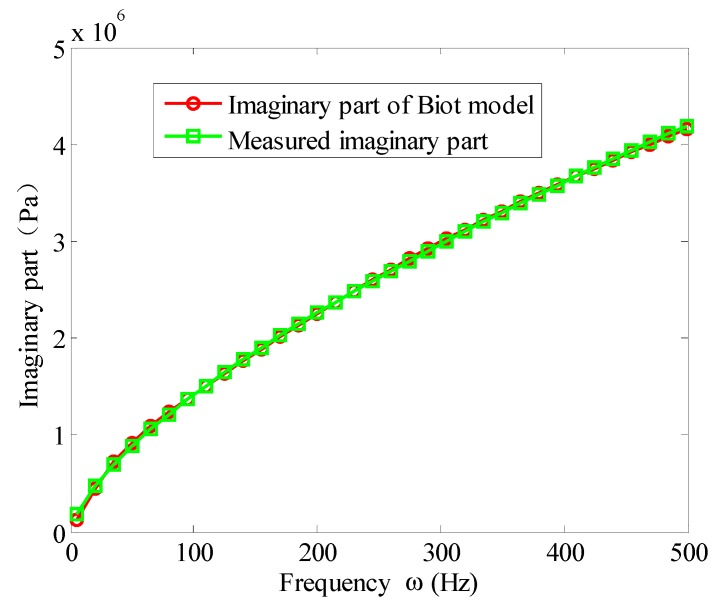
Imaginary part fitting of the Biot model.

**Figure 9 materials-12-03390-f009:**
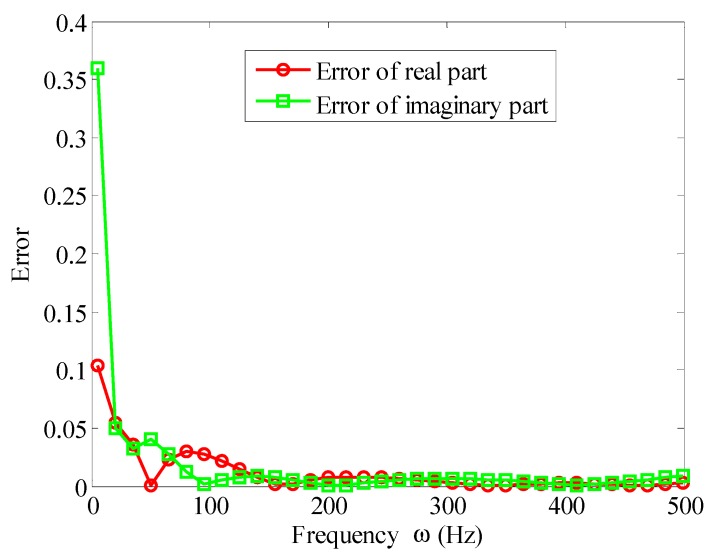
Curve-fitting relative error of the Biot model and the measured values.

**Figure 10 materials-12-03390-f010:**
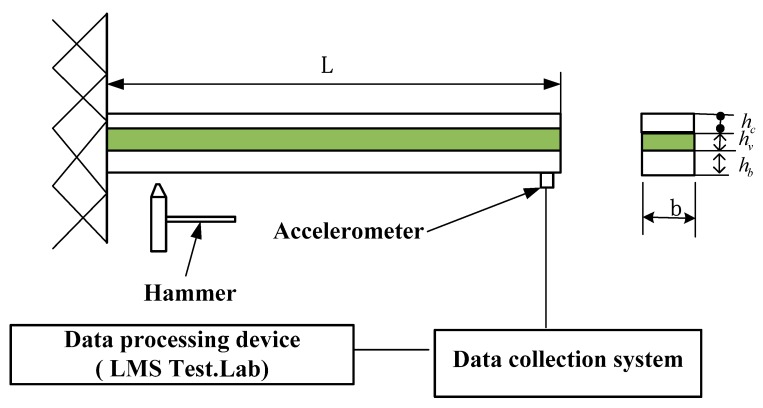
Experimental device for viscoelastic sandwich cantilever beam.

**Table 1 materials-12-03390-t001:** Measured values of storage modulus and loss factor of the viscoelastic materials at different frequencies at 30 °C.

Frequency (Hz)	5	10	30	60	100	150	200	220	240
The storage modulus (MPa)	0.51	0.62	0.91	1.1	1.43	1.71	1.92	1.73	1.76
Loss factor	0.63	0.75	0.9	1.03	1.12	1.18	1.2	1.2	1.21
Frequency (Hz)	270	300	340	360	400	440	460	500	600
The storage modulus (MPa)	1.79	1.82	1.84	1.88	1.92	2.1	2.15	2.27	3.23
Loss factor	1.2	1.22	1.21	1.21	1.21	1.21	1.22	1.22	1.23

**Table 2 materials-12-03390-t002:** Biot model parameters of the viscoelastic material, 30 °C.

Parameters	k=1	k=2	k=3
*G^∞^*	5.1e5		
*a_k_*	1.4406	4.9338	202.3130
*b_k_*	359.5605	2834.2208	114811.7290

**Table 3 materials-12-03390-t003:** Material and mechanical properties of the experimental viscoelastic sandwich beam structure.

Material Properties	Constraining Layer (Aluminum)	Base Beam(Aluminum)	Viscoelastic Layer (ZN-1 )
Elastic Modulus (GPa )	69	69	Table 2
density (kg/m^3^ )	2700	2700	1010
Poisson’s ratio	0.3	0.3	0.3
Thickness (mm)	0.78	1.91	0.40
Length (mm)	290	290	290
Width (mm)	25	25	25

**Table 4 materials-12-03390-t004:** Comparison of experimental results and finite element calculations.

Order	Experimental Result		Finite Element Model This Paper			
	Natural Frequency (Hz)	Loss Factor η	Natural Frequency (Hz)	Error (%)	Loss Factor η	Error (%)
1	24.3	0.1123	25.1	3.29	0.1152	2.58
2	151.5	0.2784	145.8	3.78	0.2910	4.53
3	390.5	0.3212	375.7	3.79	0.3362	3.87

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
