# Peer review of "A Finite Element Model for the Vibration Analysis of Sandwich Beam with Frequency-Dependent Viscoelastic Material Core"

_materials, 2019, doi:10.3390/ma12203390_

Round 1

Reviewer 1 Report

The paper presents an interesting study and deserves to be considered for publication in the journal Materials. I appreciate thoroughly conducted literature search and theoretical part of the study.

Before publication, please consider following comments that may enhance the manuscript:

Last paragraph in the Introduction part: Please define your hypothesis.

2.2. Assumptions of a sandwich beam, point no.3: Why the influence of the moment of inertia of each layer is ignored? Please explain.

Figure 5.: Please provide better figure. The small objects on the dark picture are almost not visible. Please also define the material of the test pieces in the paragraph above.

Figure 8.: What kind of Error? Please define.

3.2. Experimental validation of the sandwich composite cantilever beam: “Its material and mechanical properties are given in Table 3.” - Only properties are defined. Material has to be defined.

Author Response

Response to Reviewer 1 Comments

Dear Reviewer,

Thank you for your comments on our manuscript titled “A finite element model for the vibration analysis of sandwich beam with frequency-dependent viscoelastic material core” (ID materials-601005). We have carefully revised the manuscript and the corresponding changes have been made and highlighted by red color in the revised manuscript. A point-by-point reply to the comments is as follows.

Point 1: Last paragraph in the Introduction part: Please define your hypothesis.

Response 1: Thank you for your advice. The hypothesis is defined in the Last paragraph in the Introduction part. The derivation of the finite element equation follows the Euler-Bernoulli beam theory. It is assumed that there is no relative sliding between the layers, and the influence of the moment of inertia is not counted. Moreover, the hypotheses for a sandwich beam are defined at 2.2 in details.

2.2. Assumptions of a sandwich beam

In order to facilitate sandwich beam modeling with viscoelastic damping treatments, the following assumptions are made:

(1) The constraining layer and the base beam satisfy two hypotheses of Euler-Bernoulli beam theory, that is, the section perpendicular to the centerline of the beam is still planar after deformation (assumption of rigid cross section),and after deformation, the plane of the cross section is still perpendicular to the deformed axis.

(2) Regardless of the vertical transverse compression deformation. It is considered that the base beam layer, the damping layer, and the constraining layer have the same deflection.

(3) The influence of moment of inertia is negligible relative to bending and tensile deformation of the elastic layers and the shear deformation of the viscoelastic layer. Therefore, in order to simplify the modeling process, the moment of inertia is ignored.

(4) The layers of the materials are firmly bonded and there is no relative sliding between the layers.

Point 2: 2.2. Assumptions of a sandwich beam, point no.3: Why the influence of the moment of inertia of each layer is ignored? Please explain.

Response 2: Thank you for your comment. The influence of moment of inertia is negligible relative to bending and tensile deformation of the elastic layers and the shear deformation of the viscoelastic layer. Therefore, in order to simplify the modeling process, the moment of inertia is ignored. An explanation is added to the revised manuscript.

Point 3: Figure 5.: Please provide better figure. The small objects on the dark picture are almost not visible. Please also define the material of the test pieces in the paragraph above.

Response 3: Thank you for your comment. A clear figure is provided, and the material of the test pieces in the paragraph above is defined in the revised manuscript.

The viscoelastic material used in the experiment is ZN-1 viscoelastic material developed by China Institute of Aerospace Materials and Technology. This material is especially suitable for constructing sandwich structure and is widely used. The ZN-1 viscoelastic material is made into standard test pieces as shown in Fig. 5, which are cylinders having a diameter and a height of 20 mm. A dynamic viscoelastic spectrometer (as shown in Fig. 6, VISCOANALYSEUR VA4000 manufactured by France METRAVIB) is used to measure the storage modulus and loss factor of the viscoelastic material at different excitation frequencies. The temperature at the time of measurement is set to 30 .These data are listed in Table 1.

Figure 5. The ZN-1viscoelastic material standard test piece.

Point 4: Figure 8.: What kind of Error? Please define.

Response 4: The error is the Curve-fitting relative error of Biot model and the measured values. It is defined in the revised manuscript.

Figure 8. Curve-fitting relative error of Biot model and the measured values.

Point 5: 3.2. Experimental validation of the sandwich composite cantilever beam: “Its material and mechanical properties are given in Table 3.” - Only properties are defined. Material has to be defined.

Response 5: Sorry for our unclear statements. The materials are defined in the revised manuscript.  

Table 3. Material and mechanical properties of Experimental Viscoelastic Sandwich Beam Structure.

Material properties

Constraining layer (Aluminum)

Base beam

(Aluminum)

Viscoelastic layer

(ZN-1)

Elastic Modulus(GPa)

69

69

Table 2

density(kg/m3)

2700

2700

1010

Poisson's ratio

0.3

0.3

0.3

Thickness (mm)

0.78

1.91

0.40

Length (mm)

290

290

290

Width (mm)

25

25

25

Reviewer 2 Report

- In the section 3.2 about experimental validation, the authors do not introduce the notion of uncertainties of measurement, the notion of repeativity and reproductibility of measurements. Is it possible to introduce and verify this point?

- From numerical model point of view, which approach can be use in future works to introduce variability in the model of the authors?

- Significative works corresponding to "[23] Daya EM, Potier Ferry" can be added in the context of frequency dependent viscoelastic sandwich :

Adhikari S, Manohar C. Transient dynamics of stochastically parametered
beams. J Eng Mech 2000;126(11):1131–40.

Druesne F, Hamdaoui M, Lardeur P et Daya EM, Variability of dynamic responses of frequency dependent viscoelastic sandwich beams with material and physical properties modeled by spatial random fields, Composite Structures, 2016, 152, 316-323.

Author Response

Response to Reviewer 2 Comments

Dear Reviewer,

Thank you for your comments on our manuscript titled “A finite element model for the vibration analysis of sandwich beam with frequency-dependent viscoelastic material core” (ID materials-601005). We have carefully revised the manuscript and the corresponding changes have been made and highlighted by red color in the revised manuscript. A point-by-point reply to the comments is as follows.

Point 1:- In the section 3.2 about experimental validation, the authors do not introduce the notion of uncertainties of measurement, the notion of repeativity and reproductibility of measurements. Is it possible to introduce and verify this point?

Response 1: Thank you for your comments, the notion of uncertainties of measurement, the notion of repeativity and reproductibility of measurement are introduced and verified in the section 3.2

The uncertainties, the repeativity and reproductibility of measurements should be considered in the experimental process. The uncertainties of measurement are “the parametes that characterizes the dispersion of the values reasonably assigned to the measurements and is related to the results of the measurements". The repeativity of measurements is the degree of consistency between a series of results obtained under the same conditions using the same method and the same test material. The reproductibility of measurements is the degree of consistency between individual results obtained under different conditions, using the same method and the same test material. During the experiment, the same test piece was measured 5 times by two operators, respectively,and then the total of 10 results were averaged as the measurement results. The results showed that the uncertainties, the repeativity and reproductibility of measurements are all within an acceptable range.

Point 2:- From numerical model point of view, which approach can be use in future works to introduce variability in the model of the authors?

- Significative works corresponding to "[23] Daya EM, Potier Ferry" can be added in the context of frequency dependent viscoelastic sandwich :

Adhikari S, Manohar C. Transient dynamics of stochastically parametered beams. J Eng Mech 2000;126(11):1131–40.

Druesne F, Hamdaoui M, Lardeur P et Daya EM, Variability of dynamic responses of frequency dependent viscoelastic sandwich beams with material and physical properties modeled by spatial random fields, Composite Structures, 2016, 152, 316-323.

Response 2:Thank you for your comments. The authors add a review on the three papers in the revised manuscript. In our future works, the shell finite element of Daya can be used as a reference for the modeling of viscoelastic composite cylindrical shells, and the approachs of Adhikari and Druesne provide more ideas for considering the frequency variation characteristics of viscoelastic materials.

Chen, etc [21] established the integral finite elements for elastic-viscoelastic composite beams. Park, etc [22] compared two finite elements of viscoelastic composite plates. Daya, etc [23] established a shell finite element for viscoelastically damped sandwich structures. However, they did not consider the frequency-dependent properties of viscoelastic materials. In order to consider the frequency-variation characteristics of viscoelastic material parameters in finite element model,Adhikar, etc [24] used a stochastic finite-element formulation that employs frequency-dependent shape functions. Druesne, etc [25] used random fields to model the spatial variability of the mechanical properties of 3M ISD112 damping polymer.

[23]Daya EM, Potier-Ferry M. A shell finite element for viscoelastically damped sandwich structures. Revue Européenne des Éléments, 2002, 11(1): 39-56.

[24] Adhikari S, Manohar C. Transient dynamics of stochastically parametered beams. J Eng Mech, 2000, 126(11): 1131–1140.

[25] Druesne F, Hamdaoui M, Lardeur P et Daya EM. Variability of dynamic responses of frequency dependent viscoelastic sandwich beams with material and physical properties modeled by spatial random fields. Composite Structures, 2016, 152: 316-323.

Round 2

Reviewer 1 Report

The manuscript was carefully revised, I have no further questions. I am pleased to recommend the manuscript for publication.